# Cryptic sex in *Leishmania* depends on *SPO11* paralogs

**Carolina M. C. Catta-Preta**[1¤]*, **Vitor Luiz da Silva**[1,2,3], **Claudio Meneses**[4], **Kashinath Ghosh**[1], **David L. Sacks**[1]*

**1** Laboratory of Parasitic Diseases, National Institute of Allergy and Infectious Diseases, National Institutes of Health, Bethesda, Maryland, United States of America, **2** Department of Biochemistry, Institute of Chemistry, University of São Paulo (USP), São Paulo, Brazil, **3** Department of Chemical and Biological Sciences, São Paulo State University (UNESP), Botucatu, Brazil, **4** Laboratory of Malaria and Vector Research, Institute of Allergy and Infectious Diseases, National Institutes of Health, Bethesda, Maryland, United States of America

¤ Current address: Department of Parasitology, Institute of Biomedical Sciences, University of São Paulo, São Paulo, Brazil
* ccattapreta@usp.br (CMCC-P); dsacks@niaid.nih.gov (DLS)

**Editor:** Álvaro Acosta-Serrano, University of Notre Dame, UNITED STATES OF AMERICA

## Abstract

Genetic exchange in *Leishmania* is established, yet the molecular mechanisms enabling hybrid formation in sand flies remain poorly defined. In *Leishmania*, as in plants and several protists, two paralogs of the conserved meiotic endonuclease SPO11 are present, but their contribution to hybridization is unknown. Here, we dissect the roles of SPO11–1 and SPO11–2 during in vivo sand fly infections using targeted gene deletions, catalytically-dead mutants, expression analyses, and genome-wide characterization of hybrid progeny. We show that both SPO11 paralogs are essential for efficient hybridization: deletion of either paralog in both parents abolishes hybrid recovery, and catalytic inactivation fails to rescue mating. When only one parent lacks SPO11–1 or SPO11–2, hybrid formation is reduced in a strain-dependent manner, revealing asymmetric requirements for each paralog. Genomic analysis of hybrids from SPO11-deficient crosses reveals polyploidy and altered parental genome contributions, including unbalanced and near-balanced triploid configurations, indicating disrupted reductional processes. Together, these results establish SPO11-dependent DNA break formation as a core requirement for *Leishmania* hybridization and define distinct, strain-specific roles for the two SPO11 paralogs.

## Author summary

*Leishmania* parasites cause diseases that affect millions of people worldwide. These parasites can exchange genetic material in the sand fly vector, generating new combinations of traits, but how this process occurs remains poorly understood. In particular, it is unclear whether this exchange involves a meiosis-like program. We investigated the role of SPO11, a protein that initiates meiotic

the Creative Commons CC0 public domain dedication.

**Data availability statement:** The WGS data generated in this study have been deposited in the NCBI BioProject database under accession code PRJNA1370935. The hybridization source data and WGS detailed analysis generated in this study are provided in S1 and S2 Tables. Requests for reagents will be fulfilled by the corresponding authors. All plasmids, transgenic parasite lines in this study are available from the corresponding authors upon request and completion of MTA agreements.

**Funding:** This work was supported in part by the Intramural Research Program of the Division of Intramural Research, National Institute of Allergy and Infectious Diseases (to D.L.S.). The funders had no role in study design, data collection and analysis, decision to publish, or preparation of the manuscript.

**Competing interests:** The authors have declared that no competing interests exist.

recombination in many organisms. Although *Leishmania* encodes two SPO11 proteins, their function had not been defined. We show that both are essential for efficient hybrid formation. When SPO11 function is disrupted, hybrid formation is impaired and the few hybrids that arise display abnormal genome content, consistent with defective genome reduction. These findings show that key steps of meiotic recombination operate in *Leishmania*, despite the absence of clearly defined sexual stages. This work provides new insight into how genetic exchange occurs in this parasite and how it may contribute to its diversity.

## Introduction

Meiotic recombination between homologous chromosomes is initiated by programmed DNA double-strand breaks (DSBs) which are subsequently repaired by the formation of crossover and non-crossover recombinant DNA molecules. In addition to generating allelic variability, this repair pathway functions to establish the physical connections between the homologs required for their proper segregation at the first meiotic division. DSBs are introduced by the Spo11 protein which shows homology to the archaeal DNA topoisomerase VIA subunit (topo VIA). Spo11 was first identified in *Saccharomyces cerevisiae* as the catalytic subunit of the meiotic DNA cleavage activity, found linked to the 5' termini of the broken DNA [1]. *Spo11* is widely conserved among eukaryotes where its disruption has been shown to produce meiotic defects in yeast [2], *Drosophila [*3], *Caenorhabditis elegans* [4], flowering plants [5], and mice [6].

Across eukaryotes, *Spo11* derives from an ancient duplication of the archeal TopoVI A subunit, giving rise to two paralogous lineages, *Spo11–1* and *Spo11–2*, that were differentially retained in modern taxa [7]. Although most eukaryotes encode a single *Spo11* gene, many plants carry three homologs: one orthologous to *Spo11–1*, one orthologous to *Spo11–2* [8], and a third (*Spo11–3*) that represents a plant-specific duplication of the *Spo11–2* lineage [5,8,9]. The two meiotic Spo11 proteins form a functional heterodimer required for DSB formation, whereas Spo11–3 functions in vegetative development [8]. Notably, several early-diverging parasitic protists also encode two *Spo11* paralogs, including members of the Apicomplexa, Amoebozoa, and Kinetoplastids [7,10]. The presence of duplicated *Spo11* homologs across such distantly related eukaryotic groups underscores how ancestral meiotic modules can be conserved, repurposed, or elaborated independently across evolution.

Little is known, however, about the expression and function of *Spo11* in parasitic protists, even in those for which meiotic sex is an obligatory part of their transmission cycle. In *Plasmodium falciparum*, *Spo11* was shown to be upregulated in the late schizont stage and to be a functional ortholog of *Spo11* in yeast, as it could complement the meiosis associated sporulation defect of a *Spo11* null mutant strain of *S. cerevisiae [*11]. *Spo11* transcription has also been detected in several Amoebozoa, including *Entamoeba histolytica* and *Entamoeba invadens*, particularly under growth stress conditions and encystation [12,13]. In *Giardia intestinalis*, green fluorescent

protein fusion constructs of homologs of meiosis specific genes localize Spo11 to the nuclei of cysts, but not to the vegetative trophozoite stage [14]. By contrast, a similar reporter made in *Trypanosoma brucei* failed to detect Spo11 expression in any life cycle stage in the tsetse fly vector, including the putative meiotic stage [15]. Importantly, even for those 'sexual' protists with transcriptionally active *Spo11* and for which intra-homolog recombination has been confirmed, the requirement for *Spo11* in this process has never been demonstrated.

*Leishmania* are parasite protists (order Kinetoplastida, family Trypanosomatidae) that produce a range of skin manifestations as well as mucosal lesions and systemic, visceral disease in their human hosts. The transmission cycle of leishmanial diseases is predominantly zoonotic, involving a variety of nonhuman reservoirs, including dogs and rodents, although anthroponotic transmission occurs in specific geographic regions and parasite species. The different clinical presentations have particular *Leishmania* strain and species associations, with over 20 species pathogenic for humans. More than one billion people living in tropical and subtropical regions are at risk of contracting one or more forms of disease. *Leishmania* have a digenetic life cycle, with their vegetative growth performed by intracellular amastigotes, principally within macrophages of the vertebrate host, and by extracellular promastigotes within the digestive tract of the sand fly vector. More than 50 different sand fly species have been implicated as natural vectors. Thus, *Leishmania* displays a remarkable genetic diversity that has allowed them to adapt to a range of invertebrate and vertebrate hosts, and even to different tissue environments within the vertebrate host. The possibility that sexual reproduction might contribute to this diversity is supported by the accumulating examples of hybrid genotypes among natural isolates that infer the occurrence of genetic exchange [16]. Genetic exchange was formally demonstrated by the recovery or visualization of hybrids from sand flies experimentally co-infected with two strains of *Leishmania* bearing different drug selectable or fluorescent markers [17,18]. Based on whole-genome sequencing analyses, the allele inheritance patterns and high recombination frequencies observed in experimental hybrids provide strong evidence that the system of genetic exchange in *Leishmania* is Mendelian and involves meiosis-like sexual recombination [16]. Experimentally at least, sex is non-obligatory, relatively rare, and confined to life-cycle stages present in the sand fly midgut.

Importantly, the putative sexual cycle in the vector remains cryptic; hybridization has not been directly observed in vivo, nor have haploid gametes or a discrete stage(s) expressing meiosis- or plasmogamy-related genes been identified [19]. Thus, the precise mode of genetic exchange remains a matter of debate, and non-meiotic processes, such as parasexuality, have been proposed [20]. To provide further insights into the genetic exchange machinery, we have employed a functional genomics approach to identify genes required for *Leishmania* hybridization in vivo. Our initial surveys identified *HOP1*, a key synaptonemal complex (SC) protein involved in chromosome pairing, and *HAP2–2*, an ancestral gamete fusogen, as essential to hybridization in flies [21]. Our current studies investigate the role of *SPO11* and demonstrate that both homologs are required in each mating partner to achieve proper hybridization frequencies, while their absence permits only aberrant hybridization events characterized by unbalanced parental contributions and widespread aneuploidy.

## Results

### *SPO11–1* and *SPO11–2* demonstrate strain-specific requirements for hybrid formation in *Leishmania*

*SPO11–1* belongs to an ancient family of type IIB topoisomerase-like enzymes that catalyze the formation of programmed DNA double-strand breaks (DSBs), initiating meiotic recombination [22]. The enzyme acts by forming a transient covalent intermediate in which an active-site tyrosine attacks the phosphodiester backbone of DNA, generating a covalent 5′-phosphotyrosyl linkage. This reaction creates the DSBs required for synapsis and crossover formation. The catalytic tyrosine is therefore essential: it is the residue that mediates DNA cleavage and re-ligation through the conserved TopoVI-A fold that is shared among eukaryotic SPO11 proteins [22,23].

To assess whether this conserved mechanism operates in *Leishmania tropica*, we analyzed *SPO11–1* null mutants previously generated in two parental strains, L747 and MA37 [21]. In addition, we produced two complementation lines: (i) a re-expressor line in which the wild type *SPO11–1* coding sequence was integrated into the ribosomal SSU locus, resulting

in constitutive expression from the rRNA promoter; and (ii) a catalyticly-dead mutant ($SPO11\text{-}1^{Y160F}$) in which the active-site tyrosine at position 160 was replaced by phenylalanine. This substitution is predicted to eliminate the hydroxyl group required for nucleophilic attack on the DNA backbone while preserving the aromatic ring and overall fold of the protein [22,24], thereby abolishing catalytic activity without destabilizing the enzyme (S1C Fig). All lines were confirmed to establish midgut infections in *Lutzomyia longipalpis* at frequencies comparable to the parental controls, indicating that neither deletion nor re-expression of *SPO11–1* compromised parasite colonization in the vector (S1E Fig).

Hybrid frequencies were first assessed in control crosses between MA37 T7Cas9 eGFP-Neo and L747 T7Cas9 mCH-Sat parental lines. As expected, wild type crosses yielded a high frequency of hybrids, assessed as the frequency of infected sand flies yielding a double drug-resistant line, representing the baseline for comparison (Fig 1A and 1B). When *SPO11–1* was deleted in only one of the two parental strains, hybrid formation showed distinct outcomes depending on the background in which the gene was deleted. Crosses involving MA37 Δ*spo11–1* and wild type L747 showed a slight, non-significant reduction in hybrid frequency ($p = 0.978$), whereas deletion of *SPO11–1* in L747 nearly abolished hybrid formation ($p = 0.0147$) (Fig 1A). This pronounced asymmetry indicates that SPO11–1 function is strain dependent. Deletion of *SPO11–1* in both parental lines completely eliminated hybrid formation ($p = 0.0106$), (Fig 1A and S1 Table). Re-expression of *SPO11–1* in the L747 Δ*spo11–1* background restored hybrid recovery to wild type levels, increasing hybrid frequency by approximately 5.5-fold compared with the null mutant line ($p = 0.0258$). By contrast, re-expression of the catalytically inactive $SPO11\text{-}1^{Y160F}$ mutant failed to significantly rescue hybrid formation ($p = 0.9807$), (Fig 1A and S1 Table). The strict dependence on the conserved catalytic tyrosine supports that the DNA-cleaving mechanism initiating meiosis is conserved in *Leishmania*.

In other eukaryotes (e.g., *Arabidopsis thaliana)*, *SPO11–2* typically acts in concert with *SPO11–1* to generate double-strand breaks. We generated Δ*spo11–2* mutants in both L747 and MA37 backgrounds, and confirmed correct integration at the target locus by PCR (S1A and S1B Fig). Complementation lines were created in parallel: one re-expressor carrying the wild type *SPO11–2* integrated into the ribosomal SSU locus, and a catalytically-dead mutant ($SPO11\text{-}2^{Y100F}$) (S1D Fig). All lines maintained normal infectivity in *Lutzomyia longipalpis* sand flies (S1F Fig).

In co-infections, deletion of *SPO11–2* again led to a strain-specific decrease in hybrid formation, but the pattern was opposite that observed for *SPO11–1*. Loss of *SPO11–2* in MA37 caused a significant drop in hybrid recovery ($p < 0.0001$), whereas deletion in L747 had a modest, non-significant effect ($p = 0.265$). Crosses between both Δ*spo11–2* parental lines produced no hybrids ($p = 0.0001$) (Fig 1B and S1 Table). Re-expression of wild type *SPO11–2* in the MA37 Δ*spo11–2* background partially restored hybrid formation to wild type levels, significantly increasing recovery by approximately 35-fold compared to the *SPO11–2* null mutant ($p = 0.003$). By contrast, re-expression of the catalytically-dead $SPO11\text{-}2^{Y100F}$ mutant failed to complement the defect, yielding hybrid frequencies indistinguishable from the null mutant ($p = 0.9539$), (Fig 1B and S1 Table). These findings show that like SPO11–1, SPO11–2 relies on its conserved tyrosine residue to initiate DNA cleavage during recombination. The coexistence of two functional, non-redundant *SPO11* paralogs in *Leishmania*, a configuration retained in plants and a subset of protists but absent in animals and fungi [7], suggests an ancestral duplication where cooperative roles in meiosis were preserved.

### Strain-dependent expression of *SPO11* paralogs mirrors their differential requirement for hybridization

Having established that both *SPO11* paralogs are catalytically required for hybrid formation involving the two *L. tropica* parental strains but contribute differently depending on the strain, we examined their expression dynamics during sand fly infection to determine whether strain-specific regulation could explain this asymmetry. To assess expression in vivo, we generated endogenously tagged lines expressing N-terminal mNeonGreen fusions of *SPO11–1* and *SPO11–2* in the MA37 and L747 T7Cas9 backgrounds (S1G and S1H Fig). The tagged proteins accumulated in a perinuclear compartment rather than in the nucleus (Fig 2B and 2E). While this may reflect a biologically relevant localization, we cannot exclude potential effects of N-terminal tagging, such as interference with normal import, possibly due to disruption of

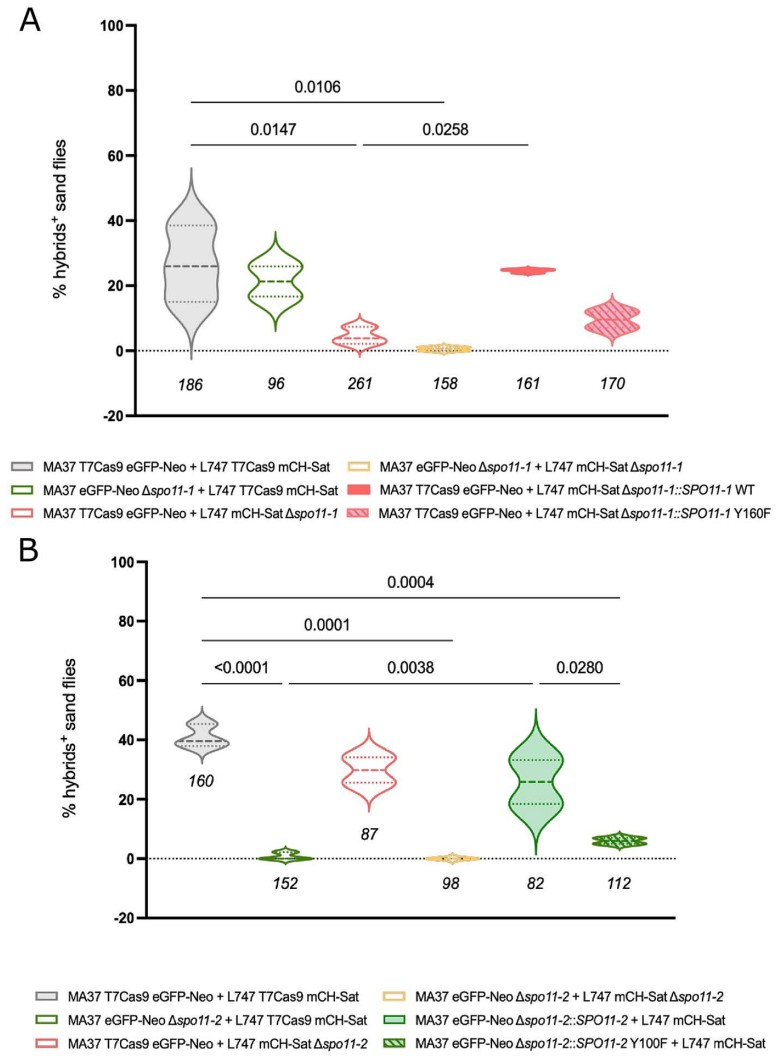

**Fig 1. *SPO11-1* and *SPO11-2* are required for hybrid formation in *L. tropica*. (A)** Percentage of sand fly midguts positive for hybrids in crosses using combinations of MA37 and L747 parental lines carrying control alleles, *SPO11-1* null mutations, or complemented lines expressing *SPO11-1*-WT or the catalytically-dead *SPO11-1*-Y106F. Violin plots represent the distribution of hybrid-positive flies across independent infections. **(B)** Percentage of sand fly midguts positive for hybrids in crosses using MA37 and L747 parental lines carrying control alleles, *SPO11-2* null mutations, or complemented lines expressing *SPO11-2*-WT or the catalytically-dead *SPO11-2*-Y100F. Violin plots represent the distribution of hybrid-positive flies across independent infections. Results are represented as the mean of at least 2 independent experiments ±SD. *p*-values were calculated by one-way ANOVA for multiple comparison of preselected pairs and a two-step step-up method of Benjamini, Krieger and Yekutieli to correct false-discovery. Sample sizes are indicated below each violin plot. Source data are provided in S1 Table.

targeting sequences near the N-terminus or alteration of protein folding. Although the N-terminal tags interfered with localization, the mNeonGreen reporters provided a reliable temporal readout of SPO11–1 and SPO11–2 expression in subpopulations of promastigotes during infection (Fig 2A and 2D). However, we cannot exclude the possibility that mistargeting may also affect protein stability or turnover. Fluorescent cells were first detected between days 3 and 5 post-infection and declined by day 8, marking a brief period of SPO11 expression that coincides with the previously defined window during which hybrids begin to be recovered from infected midguts [25]. Flow cytometry quantification of the percentage of SPO11-positive cells revealed clear strain- and paralog-specific differences. At day 5 post-infection, the

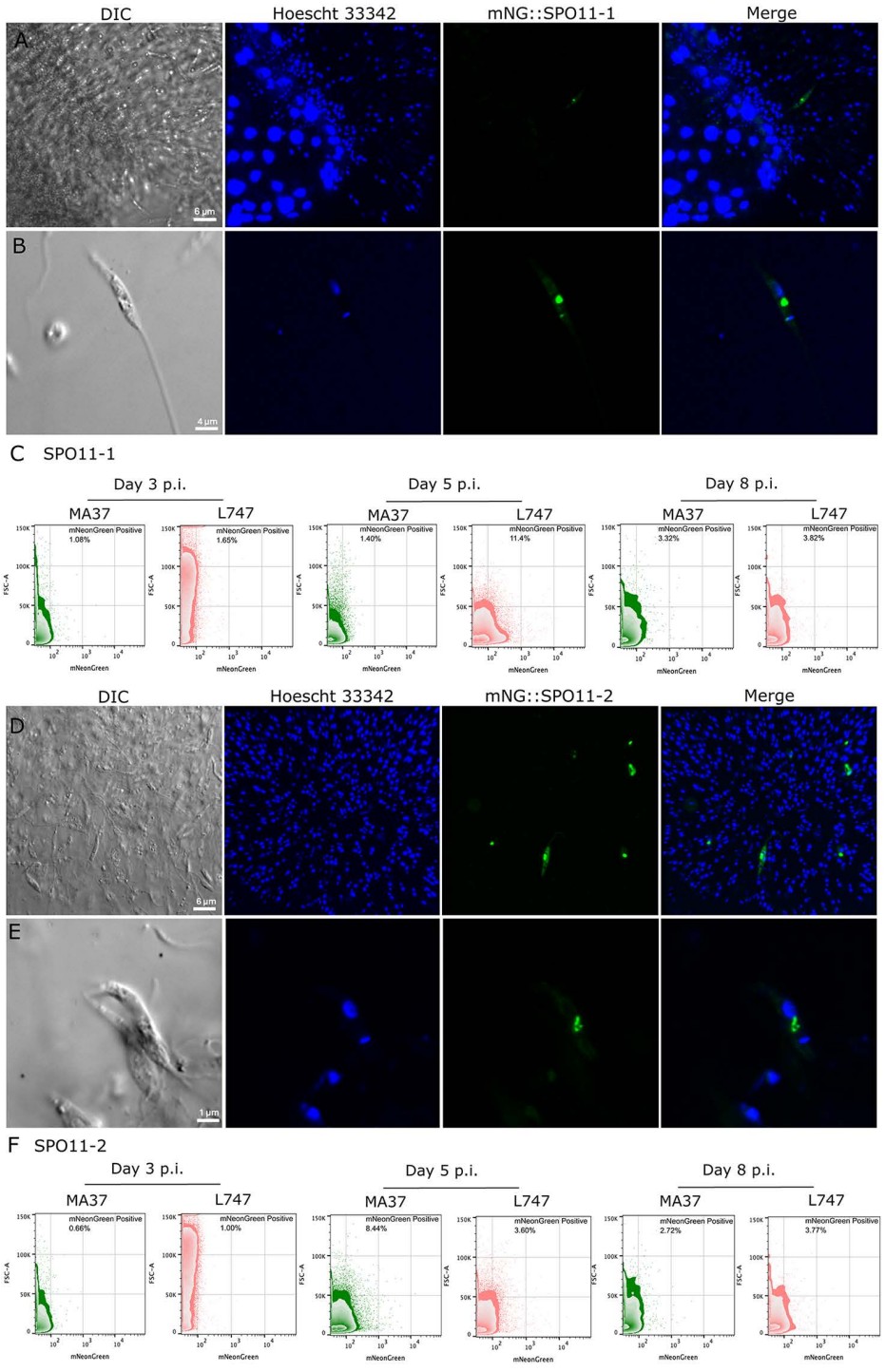

**Fig 2. SPO11 expression during sand fly infection with *L. tropica* MA37 or L747. (A)** Representative images of the anterior thoracic midgut and promastigote secretory gel (PSG) plug from sand flies infected with the MA37 mNeonGreen::SPO11-1 line at day 5 post-infection, showing a mNeonGreen-positive parasites within the plug region. **(B)** Higher-magnification view of the mNeonGreen::SPO11-1 parasites from panel A, highlighting punctate non-nuclear localization of the fluorescent signal. **(C)** Flow cytometry profiles of MA37 and L747 mNeonGreen::SPO11-1 parasites isolated from infected sand flies at days 3, 5, and 8 post-infection. **(D)** Zoom in of a PSG plug from representative sand fly infected with the MA37 mNeonGreen::SPO11-2 line at day 5 post-infection, showing mNeonGreen-positive cells in the plug region. **(E)** Higher-magnification view of MA37

mNeonGreen::SPO11-2 parasites from panel D, illustrating punctate non-nuclear localization. **(F)** Flow cytometry profiles of MA37 and L747 mNeonGreen::SPO11-2 parasites isolated from infected sand flies at days 3, 5, and 8 post-infection. Scale bars are indicated. Infections were independently repeated twice.

proportion of SPO11-positive parasites increased in both strains relative to day 3. In MA37, however, a higher proportion of parasites expressed SPO11−2 (8.44%), whereas only 3.6% expressed SPO11−2 in L747. Conversely, in L747, 11.4% of parasites were positive for SPO11−1 compared with only 1.4% in MA37. By day 8, SPO11-positive populations declined to a lower frequency in each strain (Fig 2C and 2F).

To complement these measurements, we quantified *SPO11* transcripts by RT-qPCR from day-5 midguts (S2 Fig). Unlike the reporter analyses, which measure the fraction of cells actively expressing each paralog, the RT-qPCR captures total mRNA abundance at the population level. The transcript levels did not match the protein reporter patterns. *SPO11−1* transcript abundance was nearly identical in L747 and MA37 ($p = 0.9957$), whereas *SPO11−2* transcripts were significantly lower in MA37 than in L747 ($p = 0.0300$), (S2 Fig). Within each strain, *SPO11−1* and *SPO11−2* were expressed at similar levels in L747 ($p = 0.9631$), while in MA37 *SPO11−2* showed reduced transcript abundance relative to L747 ($\log_2 FC = -0.84$, $p = 0.0300$), (S2 Fig).

Crucially, these transcript differences do not predict the protein-expression profiles, which align closely with the strain-specific functional requirements revealed in the hybridization assays. In *Leishmania*, where gene regulation is post-transcriptional, steady-state mRNA levels may fail to predict protein abundance. Accordingly, the proportion of SPO11-positive cells detected by the reporters provides the more informative readout of paralog usage in each parental background. These results support a model in which both SPO11−1 and SPO11−2 function to initiate meiotic DSB formation, but their effective contribution is shaped by the strain-specific post-transcriptional regulation.

## Loss of either *SPO11* paralog disrupts genome balance in hybrids

Analysis of hybrid populations revealed striking differences in DNA content profiles depending on the *SPO11* genotype (Figs 3 and S3). Hybrids derived from wild type parental crosses displayed predominantly diploid DNA content (89/118; 75%), with a minor proportion of near-triploid (21%) or tetraploid progeny (3%), consistent with the ploidy frequencies of experimental hybrids reported previously [21]. This pattern is consistent with meiotic division of each diploid parent and fusion of haploid gametes, with occasional hybridization between reduced and unreduced cells. By contrast, hybrids recovered from crosses involving the Δspo11−1 or Δspo11−2 parental lines were almost exclusively polyploid. Among the complete set of 74 Δspo11-derived hybrids recovered, only one was close to diploid (1.4%), while 76% were triploid and 8% were tetraploid (Fig 3A and 3B). The majority (70%) of Δspo11-derived hybrids originated from the crosses involving the respective wild type parent and either MA37 Δspo11−1 or L747 Δspo11−2, pairings that showed only a slight, non-significant reduction in hybrid frequency (Fig 1). Thus, while *SPO11* disruption of one paralog in one of the parents does not prevent hybrid formation, it compromises the fidelity of genome segregation. Disruption of *spo11−1* in L747 or *spo11−2* in MA37 not only profoundly reduced the generation of fusion competent cells, but also impaired normal reductional division. The few hybrids that were generated using these lines were either triploid or tetraploid. Re-expression of wild type *SPO11−1* in L747 Δspo11−1 or wild type *SPO11−2* in MA37 Δspo11−2 restored both hybridization competency and the expected balanced genome contributions of the hybridizing cells. Importantly, re-expression of the catalytically inactive SPO11-1^Y160F or SPO11-2^Y100F mutants failed to rescue these phenotypes (Fig 3). Together, these results demonstrate that the catalytic activity of both *SPO11* paralogs is essential for accurate genome segregation and balanced ploidy during hybridization in *L. tropica*.

## Whole-genome sequencing reveals altered parental contributions in Δspo11-derived hybrids

To investigate how loss of either *SPO11−1* or *SPO11−2* in a single parental line affects hybrid genome composition, we performed whole-genome sequencing of the parents and selected hybrids, including 1 diploid and 1 triploid hybrid

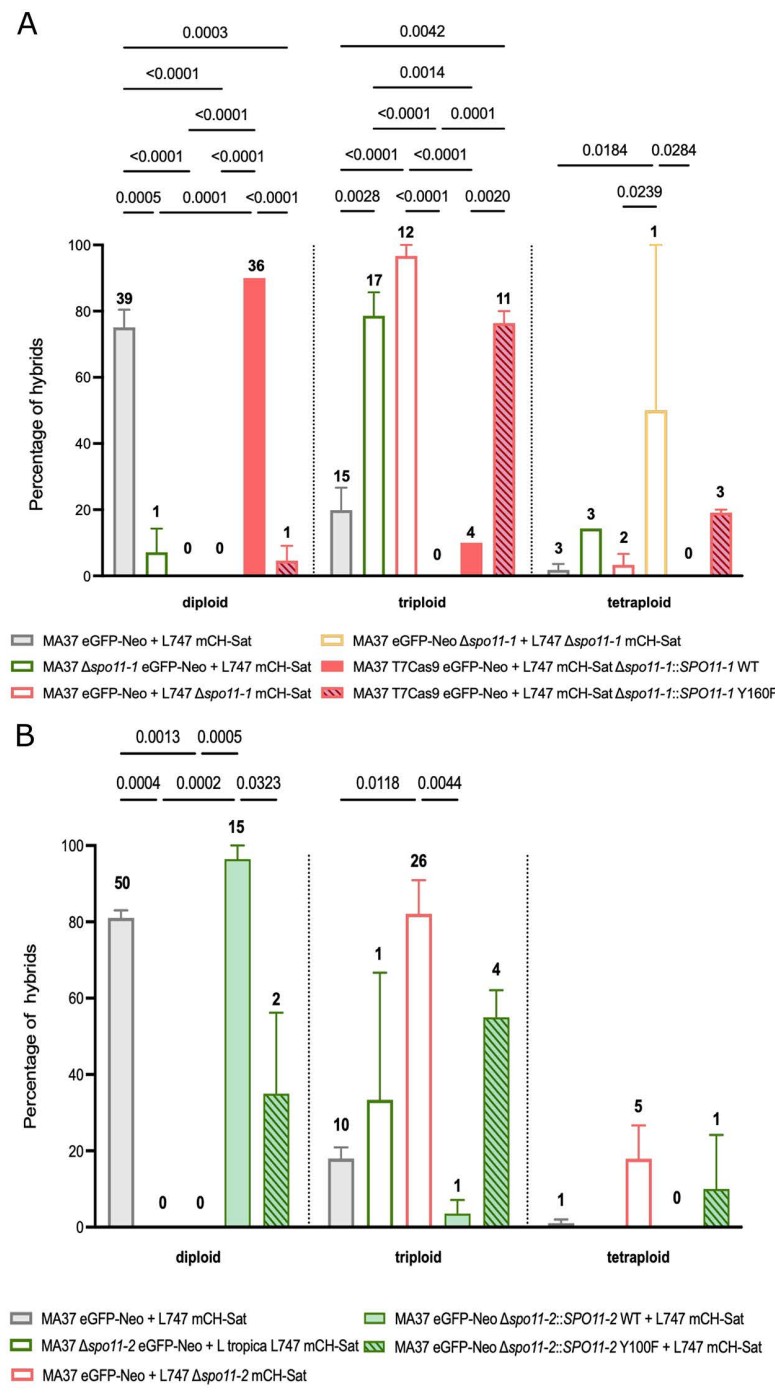

**Fig 3. Quantification of hybrid ploidy distributions in *SPO11* mutant crosses.** Quantification of the percentage of hybrids that are close to diploid (2n), triploid (3n), or tetraploid (4n) in control crosses (grey), MA37 Δ*spo11-1* eGFP-Neo × L747 mCH-Sat (green empty bar), MA37 eGFP-Neo × L747 Δ*spo11-1* mCH-Sat (red empty bar), MA37 Δ*spo11-1* eGFP-Neo × L747 Δ*spo11-1* mCH-Sat (yellow empty bar), MA37 eGFP-Neo × L747 Δ*spo11-1*::*SPO11-1* WT mCH-Sat (red filled bar), and MA37 eGFP-Neo × L747 Δ*spo11-1*::*SPO11-1* Y160F mCH-Sat (red hatched bar). Numbers above the bars represent the number of hybrids in each ploidy class for each cross. **(B)** Quantification of the percentage of hybrids that are close to diploid (2n), triploid (3n), or tetraploid (4n) in control crosses (grey), MA37 Δ*spo11-2* eGFP-Neo × L747 mCH-Sat (green empty bar), MA37 eGFP-Neo × L747 Δ*spo11-2* mCH-Sat (red empty bar), MA37 Δ*spo11-2*::*SPO11-2* WT eGFP-Neo × L747 mCH-Sat (green filled bar), and MA37 Δ*spo11-2*::*SPO11-2* Y100F eGFP-Neo × L747 mCH-Sat (green hatched bar). Numbers above the bars represent the number of hybrids in each ploidy class for each cross. **(C)** *p*-values represent statistical comparisons between the indicated groups.

generated from the wild type parents, and 8 triploid hybrids generated from crosses in which one parent carried a *SPO11–1* or *SPO11–2* deletion. Reads were aligned to the reference genome and SNPs were called across all samples. We then identified the set of homozygous SNPs that differ between L747 and MA37, and extracted allele-specific read depths at these diagnostic positions [26,27] (S2 Table).

   Genome-wide patterns of normalized read depth were first examined using a heatmap summarizing somy estimates for each chromosome across all samples and the parental contribution of all alleles (Fig 4A). Somy values deviate from exact integers, consistent with the mosaic aneuploidy that is characteristic of *Leishmania* [26] (S2 Table). The parental lines and diploid hybrid retained predominantly diploid values, while the triploid hybrid control showed ~3n values across the genome, with chromosome-specific deviations typical of *Leishmania* karyotypic plasticity. The triploid hybrids obtained from *SPO11–1* or *SPO11–2* deletions in a single parent also displayed ~3n values across the genome, again with several chromosomes deviating from the expected trisomic state. These alterations varied among hybrids, with no consistent patterns distinguishing *SPO11–1* from *SPO11–2* deletions or MA37 from L747 backgrounds.

   The diploid hybrid generated from wild type parents showed the expected equal contribution from each parent, with diagnostic SNPs reflecting a 1:1 representation of the two haplotypes across the genome. By contrast, the triploid hybrid generated from the wild type parents displayed an approximate 2:1 parental contribution across all chromosomes, with the extra genome contributed by the L747 parent. Among 8 hybrids generated from crosses in which one parent carried a *SPO11–1* or *SPO11–2* deletion, all exhibited elevated coverage consistent with triploidy. Four of these hybrids (HY029, HY030, HY032, HY033) showed a clear genome wide skew toward the *SPO11*-deficient parent. By contrast, the other hybrids analyzed, all derived from the crosses in which hybridization proceeded efficiently (HY027, HY028, HY034, HY035), presented a surprising number of trisomic chromosomes that showed near balanced parental contributions. These general patterns are illustrated by the bottle brush plots showing the parental contributions for each of the markers distributed across two representative chromosomes (Chrs 6 & 21), (Fig 4B-C). Full-genome Circos plots for all hybrids are provided in S4 Fig. Altogether, the genomic signatures in the preponderant triploid hybrids generated from *SPO11*-deficient parents supports mating between a diploid and haploid cell, and that for half of the hybrids examined, the presumed defect in meiotic reduction can be clearly assigned to the *SPO11*-deleted parent. The triploid hybrids showing more balanced parental contributions are difficult to explain, but likely reflect the same diploid-haploid cell fusion events with subsequent homogenization of chromosome copy number during vegetative growth.

## Discussion

Genetic exchange in *Leishmania* has been supported for over a decade by population genomics and experimental crosses in the sand fly vector, including the demonstration of reciprocal chromosome-internal recombination in backcross hybrids, consistent with engagement of a meiotic-like program [26]. Despite these genomic signatures, neither haploid gametes nor canonical meiotic stages have been identified, leaving the molecular details of genetic exchange in this parasite unresolved. Recent single-cell transcriptomic analyses during sand fly infection revealed low-level expression of meiotic genes such as *SPO11–1*, *SPO11–2*, and *HOP1*, distributed across multiple morphotypes rather than confined to a specialized germline-like cell type [19]. These observations indicate the presence of a meiotic toolkit but provided limited insight into which components are functionally required for recombination and genome reduction.

   A defining feature of meiotic initiation across eukaryotes is the formation of programmed DNA double-strand breaks by SPO11 enzymes, which promote homolog pairing and recombination [5]. Functional interrogation of this process has been performed extensively in animals, fungi, and plants, including *Arabidopsis thaliana*, where SPO11-mediated DSB formation is essential for homolog pairing and crossover formation, and where paralogous SPO11 proteins show partial redundancy or division of labor within a clearly defined meiotic program [8]. Functional analysis of SPO11 in protists remains limited. In *Tetrahymena thermophila*, SPO11 is required for the formation of programmed DNA double-strand breaks

PLOS Pathogens

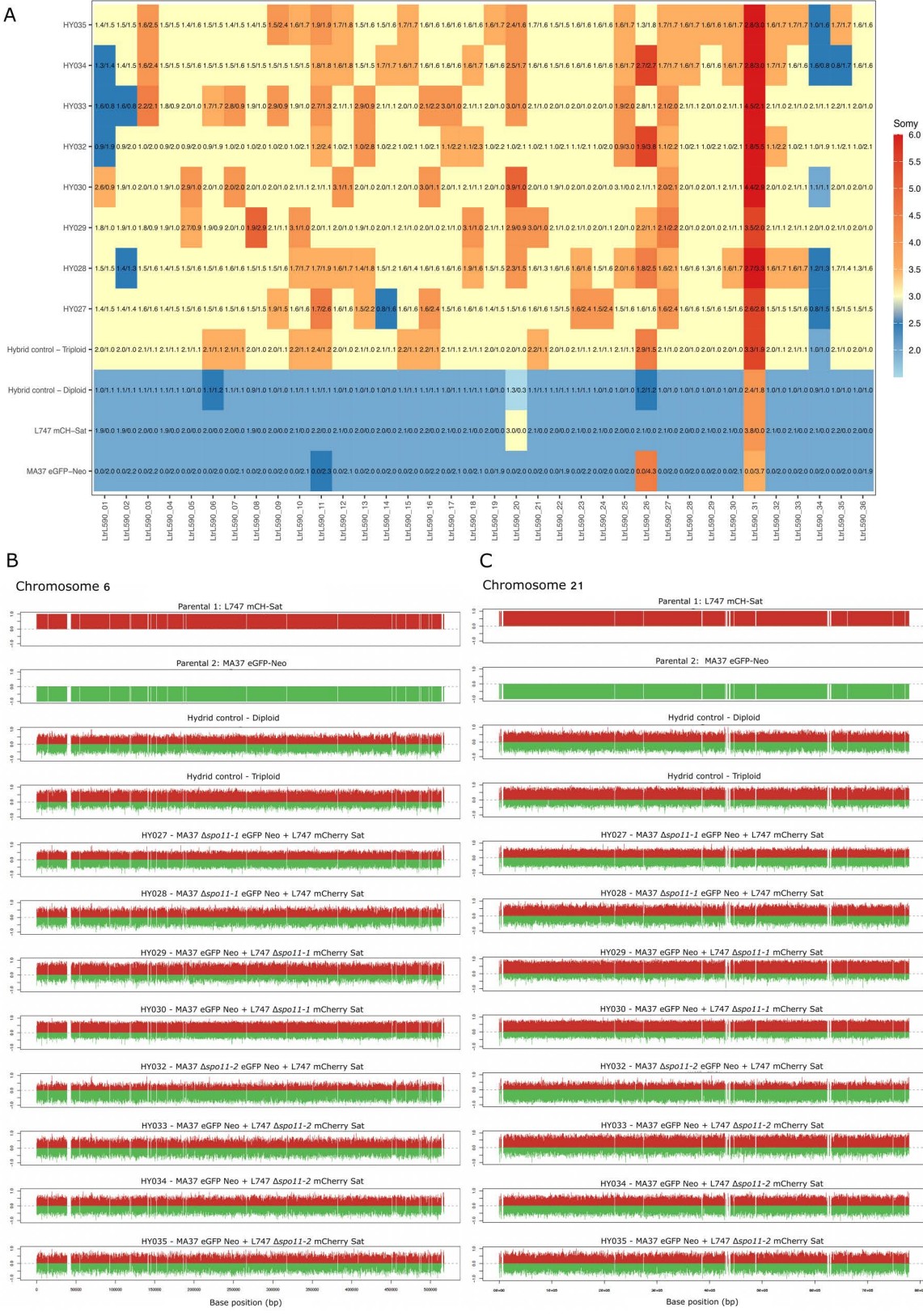

**Fig 4. Chromosome variation and parental allele contributions in *SPO11* mutant–derived hybrids.** Somy values for each chromosome are plotted as a heatmap, rounded off to the closest 0.5 value, for the two parental strains (L747 mCH-Sat and MA37 eGFP-Neo), the diploid and triploid hybrid

controls, and 8 hybrids generated from crosses involving *SPO11-1* or *SPO11-2* deletions. Overlaid are the parental inheritance values in the format L747/MA37 scaled to the nearest 0.1 value. Somy was inferred from normalized read depth across each chromosome. **(B–C)** Allele-specific read-depth ("bottle-brush") plots for representative chromosomes, Chr6 **(B)** and Chr21 **(C)**. For each hybrid clone, MA37-derived and L747-derived allele counts are shown in green and red, respectively, illustrating balanced versus skewed parental contributions.

during conjugation, a developmentally regulated sexual process that includes cytologically recognizable meiotic stages and reduction to haploid nuclei [28]. Here, we examine the role of *SPO11* paralogs in a protist with a cryptic sexual cycle, linking SPO11 activity to hybrid formation and hybrid genome architecture.

Our previous analysis of *HOP1*, a conserved HORMA-domain protein of the meiotic chromosome axis, showed that disrupting this structural component of the synaptonemal complex reduces hybrid recovery and alters hybrid ploidy [21]. The findings presented here extend our functional genomic studies to the DSB machinery itself. Both *SPO11* paralogs in *L. tropica* are essential for efficient mating, as deletion in both parents of just one paralog, either *SPO11–1* or *SPO11–2,* eliminated hybrid formation. Interestingly, deletion of one paralog in one parent also impaired hybrid formation, although in this case the requirement for either *SPO11–1* or *SPO11–2* was strain dependent. Each strain preferentially expressed one SPO11 paralog at the protein level, which in each case was observed within a defined temporal window (day 5 post-infection in the flies) that coincides with the onset of hybrid recovery [25]. Genetic deletion of the preferentially expressed paralog, leaving that parent with a single *SPO11* paralog expressed at relatively low levels, resulted in a strong hybridization defect when it was crossed with the wild type partner. By contrast, when the *SPO11* deletion left the deleted parent with the preferentially expressed paralog, hybridization frequencies were not significantly impaired. Importantly, for the crosses involving a parent with deletion of the preferentially expressed paralog, transfection with catalytically active alleles of each paralog restored hybridization while catalytically dead alleles did not, demonstrating that the conserved tyrosine-dependent DNA cleavage mechanism is indispensable for meiosis-like progression in *Leishmania*. Whether other *Leishmania* species and strains differentially express SPO11–1 and SPO11–2 is not known, and it is possible that in some genetic backgrounds their co-expression at comparable levels may reveal their functional redundancy with respect to initiation of the hybridization program.

For all but one of the 72 hybrids that were generated from crosses in which either *SPO11–1* or *SPO11–2* was deleted in one parent, their DNA content profiles indicated that they had inherited mainly triploid, or in a few progeny, tetraploid genomes. This outcome mirrors what was previously observed in crosses involving *HOP1*-deficient lines, and is consistent with a failure of one parental nucleus to undergo proper meiotic reduction prior to or during hybridization [21]. The ability of *Leishmania* to acquire fusion competency absent any apparent genome reduction is well described for cultured promastigotes submitted to stress conditions in vitro [29]. By whole genome sequencing analysis, we could determine parental contributions at the level of individual somies. For the occasional triploid hybrids that were products of wild type crosses analyzed previously, as well as the single triploid control hybrid analyzed here, 2:1 parental contributions genome wide are indicated, with the extra somy always contributed by the same parent [25,26]. In the 8 triploid, Δspo11-derived hybrids analyzed here, 4 showed 2:1 parental contribution for most chromosomes, with the *SPO11* deficient parent providing the extra genome copy, as expected. However, 4 of the hybrids showed near balanced parental contributions for the majority of their trisomic chromosomes, an outcome that is likely explained as a population-level effect of mosaic aneuploidy in which individual cells can gain or lose chromosomes during vegetative growth [20,30]. The triploid hybrid genomes appear to be exceptionally plastic, as evidenced by their increased aneuploidy, i.e., non-integral copy numbers, observed across multiple chromosomes relative to the diploid hybrid control (Fig 4A).

Altogether, the hybridization defects in the crosses involving the *SPO11* deleted parental lines reflect both a deficiency in hybrid formation, and for the cells that retain mating competency, a defect in their reductional division. Furthermore, at least for the two *L. tropica* parental strains examined, both *SPO11* paralogs are essential to generate diploid progeny with

balanced parental contributions. The genetic evidence presented here demonstrates that a core component of the meiotic machinery, SPO11-mediated DNA break formation, is functionally required for hybridization, strongly supporting the operation of a meiosis-like program. Definitive evidence for a classical meiotic cycle, including the identification of haploid gametes, will require further investigation, including the possibility that the few SPO11–1 and SPO11–2 expressing cells observed in the sand fly midgut represent haploid gametes or meiotic precursors.

## Methods

### Experimental model and details

**Ethics statement.** The mice used for blood feeding in this study were used under a study protocol approved by the NIAID Animal Care and Use Committee (protocol number LPD 68E).

***Leishmania* parasites and culture conditions.** *Leishmania tropica* strains L747 and MA37 were maintained at 26 °C in M199 medium supplemented with 10% heat-inactivated FBS, 40 mM HEPES pH 7.4, 100 µM adenosine, 1 µg/mL biotin, 5 µg/mL hemin, and antibiotics as required.

To generate null mutant cell lines, we used L747 T7Cas9-HYG mCherry-SAT and MA37 T7Cas9-HYG eGFP-NEO backgrounds, following the strategy described previously [21]. For the generation of cell lines expressing fluorescent fusion proteins, we used L747 T7Cas9-HYG and MA37 T7Cas9-HYG parental strains, as previously described [21,29].

Cultures were kept in logarithmic phase ($1–8 \times 10^6$ cells/mL) and never exceeded 20 passages from clone isolation.

**Sand flies.** *Lutzomyia longipalpis* colonies were maintained under controlled insectary conditions at 26 °C, 85% relative humidity, and a 12:12 h light–dark photoperiod, and provided 30% sucrose ad libitum. Larvae were reared on a yeast–rabbit feces substrate as previously described [31]. Female sand flies were blood-fed directly on anesthetized mice to support oviposition and colony maintenance, following the approved NIAID animal protocol (LPD 68E).

Mice were utilized both for sand fly blood feeding and, where applicable, for sand fly infection experiments. All aspects of animal use complied with the Animal Welfare Act, the PHS Policy, the U.S. Government Principles for the Utilization and Care of Vertebrate Animals Used in Testing, Research, and Training, and the NIH Guide for the Care and Use of Laboratory Animals. Animals were housed in a pathogen-free facility under a 12-hour light/dark cycle, at an ambient temperature of 22–24 °C, with relative humidity maintained at ~60%, and were provided food and water ad libitum.

### Method details

**Generation of SPO11 mutant cell lines.** For the generation of null mutants and endogenously tagged fluorescent reporters, parasites were transfected with linear repair cassettes and sgRNA templates produced according to the LeishGEdit protocol [32]. *SPO11–1* and *SPO11–2* coding sequences were also compared between L747 and MA37 and found to be highly conserved, with only a small number of amino acid substitutions and no apparent disruption of conserved motifs. Primers for sgRNAs and repair templates were designed using LeishGEdit (http://www.leishgedit.net/) based on the *L. tropica* L590 reference genome and manually curated to correct for strain-specific SNPs in L747 and MA37. Up to three mismatches per sgRNA plus homology arm were tolerated provided that no SNP occurred within the PAM site. Repair cassettes for null mutants were amplified from pTPuro_v1 templates. Repair templates for endogenous fluorescent reporters were amplified from pPLOTv1 blast-mNeonGreen-blast, using primers containing homology flanks matching the target locus. All oligonucleotides used in this paper are depicted in S3 Table.

To generate re-expressor constructs for integration into the SSU rDNA locus, we engineered a custom pLEXSY-Bsd backbone using a two-step Gibson Assembly workflow. First, the SAT resistance cassette in pLEXSY-mCherry-Sat3 (Jena Bioscience) was replaced with BSD by PCR amplifying the pLEXSY-mCherry-Sat3 vector lacking the SAT gene (forward and reverse primers designed for Gibson Assembly) and amplifying the Bsd resistance cassette from pT-BSD [32] with primers containing the appropriate overlaps. The SAT to BSD replacement was assembled using NEBuilder HiFi DNA

Assembly Master Mix (New England BioLabs), transformed into DH5alpha bacteria, and verified by colony PCR and whole plasmid sequencing.

Next, to generate the re-expression vectors for *SPO11–1* and *SPO11–2*, the mCherry ORF was removed by PCR amplification of the pLEXSY-mCH-Bsd backbone lacking mCherry. In parallel, full-length CDSs of L747 *SPO11–1* and MA37 *SPO11–2* were PCR-amplified using primers containing 20 bp overlaps to the linearized backbone. Inserts and backbone fragments were assembled by Gibson Assembly, and positive clones were identified by colony PCR using a backbone-specific primer paired with a CDS-specific primer. All constructs were sequence-validated by whole plasmid sequencing.

Final plasmids (pLEXSY-Bsd::L747*SPO11–1* and pLEXSY-Bsd::MA37SPO11–2) were prepared in midi-prep scale, linearized with SmaI (Thermo Fisher), gel-purified (QIAGEN), and used for generating re-expressor cell lines in the corresponding Δ*spo11* mutant backgrounds. To introduce the tyrosine (Y) to phenylalanine (F) point mutations into the *SPO11–1* and *SPO11–2*, site-directed mutagenesis was performed using the Q5 Site-Directed Mutagenesis Kit (New England BioLabs). Mutagenic primers were designed with the NEBaseChanger tool, positioning the desired codon substitution at the center of each primer. PCR amplification was carried out using Q5 High-Fidelity DNA polymerase, followed by KLD treatment to phosphorylate, ligate, and digest the parental plasmid. The mutagenized products were transformed into NEB 5-alpha competent *E. coli*, and correct Y to F substitutions were verified by whole plasmid sequencing. Primer sequences used for all PCR amplifications and Gibson reactions are listed in S3 Table.

Promastigotes in logarithmic phase ($1 \times 10^7$ cells/mL) were resuspended in P3 Primary Cell buffer (Lonza). For each nucleofection, 1–2 µg of total linear DNA (sgRNA cassette + repair template, or linearized SSU plasmid) was mixed with the cells immediately before transfer into a nucleocuvette. Transfections were performed using the Amaxa 4D-Nucleofector program FI-115. Parasites were recovered in 5 mL of complete M199 for 18 h before drug selection. Antibiotics were applied at the following concentrations: Pac 40 µg/mL, and Bsd 20 µg/mL. Populations were plated by limiting dilution (1:10 and 1:100) directly after selection and distributed into 96-well plates. Clones typically emerged within 10–14 days for null mutants and 7–10 days for fluorescent reporters and re-expressors.

Genomic DNA was isolated from clonal lines using the DNeasy Blood & Tissue Kit (QIAGEN). For null mutants, loss of the target CDS was confirmed using primers annealing inside the deleted gene, and proper integration of the repair cassette was confirmed by PCR from the 5′ UTR into the Pac coding region. For endogenous fluorescent reporters, correct integration was assessed using a UTR-specific forward primer and a reverse primer annealing within the reporter cassette. For re-expressor lines, SSU integration was confirmed by combining an upstream SSU primer and a reverse resistance-marker primer (BSD), and CDS-specific PCRs were performed in parallel. All primer sequences used in this study are listed in S3 Table.

## Experimental sand fly infections

All experimental infections of *Lutzomyia longipalpis* were performed under controlled insectary conditions (26 °C, 85% relative humidity, 12:12 light–dark cycle). Female sand flies aged 3–5 days post-emergence were starved for 24–30 h prior to infection. Parasites were prepared from logarithmic-phase promastigotes ($1 \times 10^7$ cells/mL), washed once in PBS, and resuspended in heat-inactivated, heparinized mouse blood. Blood feeding occurred through a chick-skin membrane stretched over glass feeders maintained at 37 °C, unless otherwise noted. Fully engorged females were separated 24 h after feeding, maintained on 30% sucrose, and dissected at the appropriate time points for parasite quantification, hybrid recovery, or imaging analyses.

To evaluate the ability of *L. tropica SPO11* mutant lines to establish midgut infections, flies were infected with either wild type, null mutant, or re-expressor parasites at $5 \times 10^6$ cells/mL in mouse blood. Flies were allowed to feed for up to 4 h and fully engorged females were separated into fresh holding cups with sucrose. Midguts were dissected at day 8 post-infection and processed for parasite count. Infections are expressed as the number of promastigotes per midgut, with at least 10 midguts counted per group.

Hybrid formation followed the strategy previously described [21]. Female flies were infected with a 1:1 mixture of two parental strains expressing distinct antibiotic markers and fluorescent proteins (e.g., L747 mCH-Sat × MA37 eGFP-Neo), with or without SPO11 deletions and re-expression, at a total parasite density of $1 \times 10^7$ cells/mL. Flies were maintained on sugar for 8 days, when individual midguts were dissected in 50 µL of M199 and homogenized to release parasites before plated directly onto complete M199 containing dual antibiotics (Neo 50 µg/mL and Sat 300 µg/mL) to select hybrids. Plates were monitored for 7–14 days, and hybrid progeny were screened by dual fluorescence, ploidy and WGS when required. The frequency of hybrid recovery was calculated as the number of hybrid-positive midguts divided by the number of dissected, parasite-positive flies.

To monitor the timing and positioning of *SPO11* positive promastigotes during *L. tropica* development in the sand fly, females were infected with SPO11–1::mNeonGreen–Bsd or SPO11–2::mNeonGreen–Bsd reporter lines at a density of $5 \times 10^6$ promastigotes/mL. Midguts were collected daily from day 3 to day 8 post-feeding, spanning the period after blood-meal digestion through anterior midgut colonization.

For fluorescence imaging, dissected midguts were briefly incubated in VectaVIEW fixation/stabilization solution (Vector Laboratories; mixed at a 1:1:1 ratio) for 20 min at room temperature. Tissues were transferred to Hoechst 33342 (10 µg/mL, Thermo Fisher) for 20 min to label nuclei. After a final PBS wash, midguts were mounted on pre-chilled glass slides in CyGEL Sustain (Abcam) to immobilize parasites, sealed with coverslips, and maintained on ice until imaging.

Confocal images were acquired on a Leica SP8 WLL FLIM microscope using a 40×/1.3 NA oil objective. Acquisition settings included bidirectional scanning, 600 Hz scan speed, line averaging (3×), and sequential frame collection to minimize channel crosstalk. Excitation was performed with the 405 nm diode laser for Hoechst and the white-light laser (70% output) tuned to excite mNeonGreen. Emission was collected at 410–450 nm (Hoechst) and 480–510 nm (mNeonGreen). Image stacks were processed in Imaris 9.8.2 (Oxford Instruments), and Z-projections were generated using identical settings across all samples.

Flow cytometry quantification of SPO11 reporter–positive parasites was performed on material recovered from pools of midguts at days 3–8 post-infection. Midguts were dissected in cold PBS, gently disrupted with a sterile pestle, and passed through a 40-µm strainer to remove debris before centrifugation at 1,500 × g for 10 min at 4 °C. Pelleted parasites were washed once in PBS and resuspended in PBS containing 2% FBS. Samples were kept on ice and analyzed immediately on a FACSCanto II and FACSDiva software (BD Biosciences). Between 50,000 and 100,000 events were collected at low flow rate for each sample, and promastigotes were identified based on FSC/SSC profiles followed by doublet exclusion using FSC-H/W and SSC-H/W parameters. SPO11–1–positive parasites were identified based on mNeonGreen fluorescence, using the background cell line signal and mock-infected dissected midguts as negative controls. Instrument settings were maintained constant across experiments, and wild type non-fluorescent parasites and mock-infected flies served as negative controls to establish the fluorescence threshold. Data were analyzed using FlowJo v10 (BD Biosciences).

## Ploidy assessment of hybrid progeny by DNA content analysis

DNA content was measured in hybrids by propidium iodide (PI) staining of fixed promastigotes followed by flow cytometry. Briefly, parasites were harvested in mid-log phase, washed once in ice-cold PBS, and fixed in 0.4% paraformaldehyde, followed by permeabilization with cold methanol added dropwise during gentle vortexing. Cells were incubated on ice for 1 h and then washed with PBS to remove residual methanol. Parasites were resuspended in PI staining buffer (PBS containing 25 µg/mL PI and 250 µg/mL RNase A) and incubated for 45 min at 37 °C in the dark. Samples were acquired on a FACSCanto II, collecting 10,000 single-cell events per sample. Doublets were excluded using FSC-H/FSC-A and SSC-H/SSC-A gating. Ploidy profiles were visualized and quantified in FlowJo v10, using WT diploid controls to anchor the 2N peak and previously validated laboratory triploid and tetraploid lines to confirm gating boundaries. Hybrid lines were classified as diploid, triploid, or tetraploid based on the position and width of the PI fluorescence intensity peaks relative to these internal standards.

 

**mRNA levels of *SPO11-1* and *SPO11-2* in *Leishmania tropica* strains by RT-qPCR**

Total RNA was obtained from whole sand flies infected with each wild type *L. tropica* strain (MA37 and L747). For each replicate, 40 infected sand flies per strain were pooled, macerated, and lysed using the Precellys 24 Touch homogenizer to extract total RNA using the RNeasy Plus Micro Kit (Qiagen) according to the manufacturer's instructions. RNA concentration and integrity were assessed using a DeNovix DS-11 Series spectrophotometer. Approximately 40 ng of total RNA was used per reaction for RT-qPCR analysis. Reverse transcription and quantitative PCR amplification were performed in a single step using the Luna Universal One-Step RT-qPCR Kit (New England Biolabs) in a final reaction volume of 20 μL per well, following the manufacturer's protocol. Reactions were run in technical triplicates. Cycle threshold (Ct) values were obtained using an Applied Biosystems QuantStudio 6 Flex Real-Time PCR System, and relative gene expression levels were calculated using the $2^{-\Delta\Delta Ct}$ method as described by Livak and Schmittgen (2001). Glyceraldehyde-3-phosphate dehydrogenase (GAPDH) was used as the endogenous reference gene to normalize expression levels of *SPO11–1* and *SPO11–2*. Primer amplification efficiency for each target gene was experimentally determined based on the slope of standard curves generated by RT-qPCR using five-fold serial dilutions of total RNA from wild type *L. tropica* L747. Primer sequences and primer efficiency are provided in S3 Table and S2 Fig, respectively.

**Whole-genome sequencing (WGS)**

Genomic DNA from clonal parental lines and sand fly–derived hybrid progeny was purified using the DNeasy Blood and Tissue Kit (QIAGEN) and submitted to Psomagen (Rockville, MD) for whole-genome sequencing. DNA libraries were prepared using the TruSeq Nano DNA Library Prep Kit (Illumina), and 150-bp paired-end reads were generated on an Illumina NovaSeq X platform. Sequencing reads were aligned to the *L. tropica* Ltr590 v68 reference genome (TriTrypDB) using BWA-MEM v0.7.17 with default settings. Sequencing quality and mapping statistics, including mean mapped read coverage for each sample, were obtained from Qualimap v2.2.1. All downstream analyses, including Single nucleotide polymorphisms (SNPs) and inheritance patterns were characterized using the PAINT software suite, which is optimized for analyzing mosaic and aneuploid *Leishmania* genomes.

PAINT was used to identify homozygous marker differences between the two parental strains and to compute chromosome somies based on normalized median read depth in 5-kb windows using the *ConcatenatedPloidyMatrix* module. Regions affected by high copy number variation or repetitive sequence were excluded by removing windows with coverage ≥2× or ≤0.5×the chromosome average. For polyploid hybrids (≥3n), estimated somies derived from read depth were adjusted by dividing values by 2 and multiplying by the ploidy determined independently by DNA content analysis (PI staining) and the expected parental contribution ratio (e.g., 1:1 or 2:1).

SNP calls with allele frequencies between 0.15–0.85 were considered heterozygous, whereas positions with allele frequencies >0.85 were designated as homozygous. Variants supported by <10 total reads, <25% representation in either read direction, or with allele frequencies <0.15 were excluded from further analysis. The parental origin of each homozygous SNP in hybrid genomes was assigned using the *getParentAlleleFrequencies* function in PAINT. Parental allele frequencies were exported in a format compatible with Circos v0.69, and inheritance circos plots were generated using homozygous marker SNPs distinguishing MA37-GFP (green) from L747-mCherry (red) parental backgrounds.

**Statistical analysis**

Hybridization experiments were repeated at least twice per group. No randomization or blinding procedures were used in this study. For statistical analysis, data was plotted and analyzed on GraphPad Prism 10.3. We used Ordinary one-way ANOVA for multiple comparison of preselected pairs and a two-step step-up method of Benjamini, Krieger and Yekutieli to correct false discovery. *p*-values <0.05 were considered significant.

## Supporting information

**S1 Fig. Validation of *SPO11–1* and *SPO11–2* mutant and complemented lines, and impact on parasite load in sand flies.** (A–B) PCR confirmation of *SPO11–2* deletion in MA37 (A) and L747 (B) clones using CDS-, UTR- and PAC-specific primers. (C–D) PCR genotyping of L747 Δ*spo11–1*::SPO11–1-WT and Δ*spo11–1*::*SPO11–1*-Y160F (C), and MA37 Δ*spo11–2*::SPO11–2-WT and Δspo11–2::*SPO11–2*-Y100F (D), confirming integration into the SSU loci. (E–F) Total number of promastigotes per midgut at day 8 post-infection in flies infected with *SPO11–1* lines (E) or *SPO11–2* lines (F), including parental controls, null mutants, and complemented lines. Each point represents one midgut; horizontal bars show median values. "ns" indicates non-significant comparisons. (G–H) PCR validation of mNeonGreen tagging of *SPO11–1* (G) and *SPO11–2* (H) in MA37 and L747 clones. WT = wild type; Cl1 and Cl2 = independent tagged clones. (PDF)

**S2 Fig. Primer efficiency and RT–qPCR analysis of *SPO11–1* and *SPO11–2* expression in *L. tropica*.** (A) Primer amplification efficiency determined from standard curves generated by five-fold serial dilutions of total RNA for *GAPDH* (endogenous control), *SPO11–1*, and *SPO11–2*. (B) Representative melting curves for *GAPDH*, *SPO11–1*, and *SPO11–2*, demonstrating single, specific amplification products. (C) Standard curves generated from five-fold serial dilutions of total *L. tropica* RNA and used to calculate primer amplification efficiency for *GAPDH* (endogenous control), *SPO11–1*, and *SPO11–2*. Linear regression lines, slopes, and coefficients of determination ($R^2$) are shown. (D) Amplification plots for experimental samples. (E) Corresponding melting curves for *GAPDH*, *SPO11–1*, and *SPO11–2* from experimental RT–qPCR reactions performed on infected whole flies. (F) Relative mRNA expression levels of *SPO11–1* and *SPO11–2* in pools of 40 *Lu. longipalpis* infected with either the MA37 or L747 *L. tropica* strain, measured by RT–qPCR. Bars represent the mean of three independent biological replicates, with individual data points overlaid; values were normalized to *GAPDH*. (G) $\log_2$ fold change in *SPO11–1* and *SPO11–2* expression in L747 relative to MA37, calculated from the RT–qPCR data shown in panel G. Values represent means from three independent experiments. (PDF)

**S3 Fig. DNA content–based classification of hybrid ploidy by propidium iodide (PI) staining.** Representative flow cytometry histograms of propidium iodide (PI)–stained parasites illustrating how hybrid DNA content was classified. Diploid, triploid, and tetraploid control lines were used to define reference peaks corresponding to 2n, 3n, and 4n DNA content, respectively. Hybrid populations were classified by comparison to these controls. Multiple representative profiles are shown for triploid hybrids, illustrating the range of DNA content distributions observed among independently derived hybrid clones recovered from SPO11-deficient crosses. (PDF)

**S4 Fig. Genome-wide parental allele contributions in *SPO11* mutant–derived hybrids.** Circos plots showing allele-specific read-depth profiles across all 36 *Leishmania tropica* chromosomes for the two parental strains (L747 mCH-Sat and MA37 eGFP-Neo), the diploid and triploid hybrid controls, and hybrids generated from crosses involving *SPO11–1* (A) or *SPO11–2* (B) deletions. MA37-derived and L747-derived allele counts are shown in green and red, respectively. Balanced hybrids display near-equal parental contributions across most chromosomes, whereas unbalanced hybrids show genome-wide skewing toward the *SPO11*-deficient parent. (PDF)

**S1 Table. Summary of all *L. tropica* crosses performed in this study.** (XLSX)

**S2 Table. Whole-genome sequencing summary and analysis of parental lines and hybrids.** (XLSX)

**S3 Table. Oligonucleotides used in this study.**
(XLSX)

## Acknowledgments

We thank all members of the Sacks lab for discussions. We thank Dr. Shehre-Banoo Malik for drawing our attention to *Leishmania SPO11–2*.

## Author contributions

**Conceptualization:** Carolina M. C. Catta-Preta, David L. Sacks.

**Formal analysis:** Carolina M. C. Catta-Preta, Vitor Luiz Da Silva.

**Funding acquisition:** David L. Sacks.

**Investigation:** Carolina M. C. Catta-Preta, Vitor Luiz Da Silva.

**Methodology:** Carolina M. C. Catta-Preta.

**Resources:** Carolina M. C. Catta-Preta, Claudio Meneses, Kashinath Ghosh, David L. Sacks.

**Supervision:** David L. Sacks.

**Writing – original draft:** Carolina M. C. Catta-Preta.

**Writing – review & editing:** Carolina M. C. Catta-Preta, David L. Sacks.

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
