## [Decision Letter · Decision Letter 0]

2 Apr 2026

PPATHOGENS-D-26-00360

Cryptic Sex in Leishmania Depends on SPO11 Paralogs

PLOS Pathogens

Dear Dr. Catta-Preta,

Thank you for submitting your manuscript to PLOS Pathogens. After careful consideration, we feel that it has merit but does not fully meet PLOS Pathogens's publication criteria as it currently stands. Therefore, we invite you to submit a revised version of the manuscript that addresses the points raised during the review process.

We look forward to receiving your revised manuscript.

Kind regards,

Álvaro Acosta-Serrano

Academic Editor

PLOS Pathogens

Margaret Phillips

Section Editor

PLOS Pathogens

Sumita Bhaduri-McIntosh

Editor-in-Chief

PLOS Pathogens

orcid.org/0000-0003-2946-9497

Michael Malim

Editor-in-Chief

PLOS Pathogens

orcid.org/0000-0002-7699-2064

**Additional Editor Comments:**

As you will see from the reports, the reviewers are enthusiastic about the work. We invite you to submit a revised version of the manuscript addressing the minor comments raised.

**Journal Requirements:**

At this stage, the following Authors/Authors require contributions: Carolina Moura Costa Catta-Preta, Vitor Luiz Da Silva, Claudio Meneses, Kashinath Ghosh, and David L. Sacks. Please ensure that the full contributions of each author are acknowledged in the "Add/Edit/Remove Authors" section of our submission form.

https://journals.plos.org/plospathogens/s/submission-guidelines#loc-parts-of-a-submission

- ® on page: 12 and 15.

**Reviewers' Comments:**

Reviewer's Responses to Questions

**Part I - Summary**

Reviewer #1: Very little is known about the mechanism of meiosis in kinetoplastids, which are evolutionarily divergent from commonly studied model eukaryotes such as yeast and humans. For example, in Leishmania, formation of a gamete with a haploid genome has not been observed. This manuscript by Catta-Preta and colleagues reports characterization of two paralogs of a key meiotic recombination protein SPO11 (SPO11-1 and SPO11-2) for hybridization in Leishmania tropica (using L747 and MA37 parental strains that allow SNP analysis). By analyzing hybridization frequencies and performing whole genome sequencing of hybrids that derived from the parents whose SPO11-1 or SPO11-2 genes are deleted, they demonstrate the importance of the catalytic activity of these SPO11 proteins. Importantly, their somy analysis showed a genome-wide skew toward the SPO11-deficient parent, which could be interpreted as an indirect evidence for SPO11-dependent haploidization in Leishmania.

Overall the manuscript is well written and structures well presented. A previous paper by these authors published in Nature communications (2023) used similar techniques to demonstrate the importance for other key meiotic factors (HOP1 and HAP2) in hybridization. In this sense, one may feel that the progress reported in the current manuscript is rather incremental. However, given how little is known about the meiotic mechanism in Leishmania and kinetoplastids in general, I would like to strongly support the publication of this manuscript in Plos Pathogens. Below are minor comments, which the authors may consider to address in the revised manuscript.

- Are the SPO11-1 and SPO11-2 protein and mRNA (coding and UTRs) sequences identical in the L747 and MA37 lines?

- Line 183, perinuclear compartment rather than in the nucleus: Do the authors know that the mNG-tagged proteins are non-functional? I have an impression that this localization pattern is actually similar to what the authors observed for HOP1 in their 2023 paper (Figure 6). Could the obtained images represent a genuine localization of SPO11 proteins?

Bungo Akiyoshi

Reviewer #2: This is an interesting paper. The existence of sex in Leishmania has been known, or inferred, for many years. However, the mechanisms involved are only slowly being dissected. The role of the process in generating parasite diversity has possible implications stretching from disease pathogenesis to drug resistance. This paper focuses on the role of the meiotic endonuclease SPO11, a widely conserved protein which has a central role in generating the double stranded DNA breaks that are required to initiate recombination. Overall, the paper is well written, the experiments are competently performed, and the conclusions are supported by the data. In preparing a final version of the manuscript, there are a few points which the authors should clarify or comment on, as set out below.

Reviewer #3: The authors have made key contributions to our understanding of cryptic sex in Leishmania and in the presented study they provide a clear and rigorous demonstration of the importance of the conserved meiotic endonuclease SPO11, present in this parasite as two paralogs, SPO11-1 and SPO11-2. The roles of these paralogs are investigated in in vivo sandfly infections using a well-integrated experimental strategy that includes gene expression analysis with tagged versions of the proteins, gene knockouts with complementation by wild-type and catalytically inactive variants, and genome-wide analysis of hybrid progeny. The study is methodologically robust and logically coherent. The data convincingly establish that SPO11-dependent DNA double-strand break formation is central to Leishmania hybridization, while also revealing strain-specific functional differences between the paralogs.

Given that the sexual cycle in the vector remains cryptic, without direct observation of hybridization, haploid gametes, or defined meiotic stages, and that alternative mechanisms such as parasexuality have been proposed, this work provides important mechanistic insight. Building on prior identification of factors involved in chromosome pairing and hybrid formation, the authors demonstrate that both SPO11 paralogs are required in each mating partner for efficient hybridization. Loss of either paralog leads to aberrant hybrids with unbalanced parental contributions and widespread aneuploidy.

**Part II – Major Issues: Key Experiments Required for Acceptance**

Reviewer #1: (No Response)

Reviewer #2: NA

Reviewer #3: The study is methodologically and conceptually robust, with no major issues identified.

**Part III – Minor Issues: Editorial and Data Presentation Modifications**

Reviewer #1: (No Response)

Reviewer #2: 1. In Fig. 1, which describes the extent of hybrid formation, the violin plots in (A) and (B) of the control cross (MA37 T7Cas9 eGFP-Neo + L747 T7Cas9 mCH-Sat) display a different profile. Are these differences significant?

2. The manuscript reports that N-terminal tagging of SPO11-1 and SPO11-2 apparently disrupts nuclear targeting of the proteins, but concludes that the pattern of expression provides a reliable temporal readout of the process. This is probably a reasonable assumption, but the authors should mention the caveat that they cannot exclude the possibility that mis-targeting may have an impact on the extent of protein turnover. It would also be interesting to hear the author’s views on why only a small percentage of the parasites express detectable protein, and what the mechanism(s) for this might be. This could be mentioned at the appropriate place in the Results, or in the Discussion.

3. The differences in DNA content of the hybrid populations is intriguing. Have the authors checked on the infectivity of these parasite lines, and do they know if there is an impact on genome content as a result of continuous rounds of macrophage infection in vitro, or even as a result of experimental mouse infections. It would be nice to see some information on infectivity, but it is probably unreasonable to insist on genome stability data, if they are not available.

Reviewer #3: Introduction:

Page 2 – lines 61- 63. For this sentence, it would be helpful to provide a reference to allow readers to easily access the article.

Page 3 – line 83 – “The transmission cycle of leishmanial diseases can be anthroponotic or zoonotic”. While this is correct, the statement may be misleading, as leishmaniasis is predominantly zoonotic, with anthroponotic transmission limited to specific geographic regions and parasite species. I suggest clarifying or rephrasing this sentence to better reflect this distinction.

Page 3 – lines 85-86: “The different clinical presentations have particular Leishmania strain and species associations, with over 20 species pathogenic for humans.” Here again, the issue is one of emphasis. I recommend focusing on species rather than strains, as species-level differences are the primary determinants of clinical manifestations. Accordingly, “strains” could be omitted to improve clarity and accuracy.

Page 6, second paragraph (lines 197–205): there appear to be two incorrect figure references. Where the text cites Figure S3, it likely should be Figure S2; similarly, the reference to Figure 3A also appears to be incorrect and should be revised accordingly.

PLOS authors have the option to publish the peer review history of their article (what does this mean?). If published, this will include your full peer review and any attached files.

Reviewer #1: **Yes:** Bungo Akiyoshi

Reviewer #2: No

Reviewer #3: No

**Figure resubmission:**
---

## [Editor Report · Decision Letter 1]

18 Apr 2026

Dear Dr. Catta-Preta,

We are pleased to inform you that your manuscript 'Cryptic Sex in Leishmania Depends on SPO11 Paralogs' has been provisionally accepted for publication in PLOS Pathogens.

Best regards,

Álvaro Acosta-Serrano

Academic Editor

PLOS Pathogens

Margaret Phillips

Section Editor

PLOS Pathogens

Sumita Bhaduri-McIntosh

Editor-in-Chief

PLOS Pathogens

orcid.org/0000-0003-2946-9497

Michael Malim

Editor-in-Chief

PLOS Pathogens

orcid.org/0000-0002-7699-2064
---

## [Editor Report · Acceptance letter]

Dear Dr. Catta-Preta,

We are delighted to inform you that your manuscript, "Cryptic Sex in Leishmania Depends on SPO11 Paralogs," has been formally accepted for publication in PLOS Pathogens.

Best regards,

Sumita Bhaduri-McIntosh

Editor-in-Chief

PLOS Pathogens

orcid.org/0000-0003-2946-9497

Michael Malim

Editor-in-Chief

PLOS Pathogens

orcid.org/0000-0002-7699-2064